

# Emissions of Intermediate- and Semi-Volatile Organic Compounds (I/SVOCs) from Different Cumulative Mileage Diesel Vehicles under Various Ambient Temperatures

Shuwen Guo[1], Xuan Zheng[1*], Xiao He[1], Lewei Zeng[1], Liqiang He[2], Xian Wu[2], Yifei Dai[3], Zihao Huang[1],

Ting Chen[4], Shupei Xiao[1], Yan You[5], Sheng Xiang[6], Shaojun Zhang[4], Jingkun Jiang[4], and Ye Wu[4]

[1]College of Chemistry and Environmental Engineering, Shenzhen University, Shenzhen 518060, China

[2]State Environmental Protection Key Laboratory of Vehicle Emission Control and Simulation, Chinese Research Academy of Environmental Sciences, Beijing 100012, China

[3]Institute for Advanced Study, Shenzhen University, Shenzhen 518060, China

[4]School of Environment, State Key Joint Laboratory of Environment Simulation and Pollution Control, Tsinghua University, Beijing 100084, China

[5]National Observation and Research Station of Coastal Ecological Environments in Macao, Macao Environmental Research Institute, Macau University of Science and Technology, Macao SAR 999078, China

[6]State Key Laboratory of Pollution Control and Resource Reuse, Tongji University, Shanghai, China 200092, China

*Correspondence to*: Xuan Zheng (x-zheng11@szu.edu.cn)

**Abstract.** The role of intermediate- and semi-volatile organic compounds (I/SVOCs) in heavy-duty diesel vehicle (HDDV) exhaust remains a significant research gap across previous studies, with limited focus on cumulative mileage and ambient temperature effects. This study analyzed gaseous and particulate I/SVOCs from four in-use HDDVs using thermal desorption two-dimensional gas chromatography-mass spectrometry (TD-GC×GC-MS). Total I/SVOC emission factors (EFs) ranged

from 9 to 406 mg·km$^{-1}$, with 79 – 99 % in the gaseous phase. High-mileage vehicles (HMVs) emitted I/SVOCs at levels eight times greater than low-mileage vehicles (LMVs), highlighting the influence of cumulative mileage. Emission deterioration occurred under both cold-start and hot-running conditions, though HMVs showed no extra sensitivity to cold starts. HMVs also exhibited increasing emissions with component volatility, alongside a higher share of oxygenated I/SVOCs (O-I/SVOC) than LMVs (65% vs. 42%). Compounds such as phenol, alkenes, and cycloalkanes appeared only in HMV emissions.

Temperature effects were notable at 0°C, only HMV emissions rose significantly, while LMV emissions remained stable. A strong linear correlation ($R^2$ = 0.93) between I/SVOC EFs and modified combustion efficiency (MCE) suggests that reduced combustion efficiency drives higher I/SVOC emissions. HMVs also showed four times greater secondary organic aerosol formation potential (SOAFP) compared to LMVs. This increase was smaller than the eightfold rise in EFs, likely due to the higher O-I/SVOC content in HMV emissions.


**Keywords.** HDDVs, I/SVOCs, emission deterioration, cumulative mileage, ambient temperature, combustion efficiency





## 1 Introduction

As a major global air pollutant, fine particulate matter (PM$_{2.5}$) leads to over three million premature deaths yearly (Apte et al., 2018), mainly associated with lung cancer, ischemic heart disease, and stroke (Guan et al., 2018; Xue et al., 2021). Source apportionment studies indicated that secondary organic aerosol (SOA) accounts for 12% to 77% of the total PM$_{2.5}$ mass (Huang et al., 2014; Sun et al., 2020; Zhang et al., 2021). Observation studies have demonstrated that SOA contributions increase with the severity of pollution during haze episodes (He et al., 2020; Ho, 2016; Li et al., 2015). Intermediate-volatility and semi-

volatile organic compounds (I/SVOCs), with effective saturation concentrations (C*) between $10^3$ to $10^6$ and 1 to $10^2$ μg·m$^{-3}$, respectively, have been proven to be more effective SOA precursors than volatile organic compounds (VOCs), as evidenced by numerous laboratory experiments (Daniel S. Tkacik et al., 2012; Jathar et al., 2013; Morino et al., 2022; Presto et al., 2009; Zhao et al., 2014).

Heavy-duty diesel vehicles (HDDVs) are recognized as significant sources of I/SVOCs (Alam et al., 2018; Drozd et al., 2021;

Liu et al., 2021; Lu et al., 2018; Presto et al., 2009; Zhao et al., 2015). However, the contribution of HDDVs to I/SVOC emissions from on-road motor vehicles in China remains a contentious topic, as reflected in differing studies. For example, Zhao et al. (2022) reported that diesel vehicles contributed 85% of IVOC emissions from on-road mobile sources in China, while Chang et al. (2022) found that diesel vehicles emitted only about 20% of the IVOCs produced by gasoline vehicles. These variations underscore the need for a more precise assessment of diesel vehicle I/SVOC emission factors (EFs). Previous

studies have explored the impact of emission standards, after-treatment technologies, and driving cycles on EFs (Zhao et al., 2015; He et al., 2022b, a; Zhang et al., 2024a), while rare studies reported the discrepancy of I/SVOC emissions between different cumulative mileage HDDVs. Furthermore, many regions in China experience temperatures of 0°C or lower during the autumn and winter. Consequently, HDDVs operating under such low-temperature conditions may exhibit different emission characteristics compared to those under normal temperatures (e.g. 23°C). This underscores the importance of examining the

variations in I/SVOC emissions and exhaust component distribution from HDDVs across different temperature conditions.

Additionally, the complexity of I/SVOC components also impedes the accuracy of HDDV EFs. The alkanes, alkenes, alkynes, cycloalkanes, monocyclic aromatic compounds, and oxygenated organic compounds present in IVOCs are all significant precursors for the formation of SOA. Previous analyses of I/SVOCs primarily relied on traditional one-dimensional gas chromatography coupled with mass spectrometry (GC-MS). Due to limitations in separation techniques, many challenging-to-

analyze I/SVOCs have been collectively classified as an unresolved complex mixture (UCM), allowing only for rough quantification (Liu et al., 2021; Qi et al., 2019, 2021; Tang et al., 2021; Zhao et al., 2014, 2015, 2016). In the study of Zhao et al (2015), approximately 80% of I/SVOCs emitted by diesel vehicles remain unresolved by GC-MS, reckoned as UCM. Moreover, due to the variability in the response signals detected by mass spectrometry for different complex organic compounds (He et al., 2022b), the lack of detailed component information introduces significant uncertainties in I/SVOC

quantification and prediction of SOA formation potential (SOAFP). To reduce the UCM of I/SVOCs from HDDVs and reduce both qualitative and quantitative uncertainties, He et al. (2022b, a) employed comprehensive two-dimensional gas chromatography (GC×GC), which enhances selectivity, peak capacity, and sensitivity by connection of two capillary columns with complementary stationary phases in series. The authors developed a method by constructing class-screening programs based on their characteristic fragments and mass spectrum patterns to identify thousands of compounds of one GC×GC profile,

which successfully identified over 85% of I/SVOCs from HDDV exhaust (He et al., 2022b). Furthermore, they were the first to quantify oxygenated I/SVOCs (O-I/SVOCs) and find the identified O-I/SVOCs will result in a 45% difference in the prediction results of SOAFP (He et al., 2022b). Therefore, the application of GC×GC and the qualitative method based on the



unique mass spectrum patterns for I/SVOC (He et al., 2022b, a, 2024), provides a more accurate determination of I/SVOC EFs

and component distribution. This methodology offers a robust foundation for analyzing the effects of cumulative mileage and ambient temperatures on HDDV I/SVOC emissions.

In this study, a thermal desorption two-dimensional gas chromatography and mass spectrometry (TD-GC×GC-MS) was utilized to measure gaseous and particulate I/SVOCs emitted from four HDDVs based on chassis dynamometer emission tests. The I/SVOC EFs and gas-to-particle partitioning of vehicles with varying cumulative mileages under different ambient temperatures have been reported. Detailed species of I/SVOCs and their volatilities have also been reported, with an analysis

of the causes of high I/SVOC emissions from the perspective of combustion efficiency. Additionally, the generation of SOAFP from I/SVOCs in the exhaust of different vehicles was evaluated. The impact of cumulative mileage and ambient temperature on total I/SVOC emissions in the emission inventory was assessed after incorporating these factors into the EFs.

## 2 Materials and methods

### 2.1 Fleet and dynamometer tests

Four in-use HDDVs using China VI 0# diesel fuel were tested on a chassis dynamometer following China heavy-duty commercial vehicle test cycle for tractor trailers (CHTC-TT) at the China Automotive Technology & Research Center (CATARC) in Guangzhou, China. All tested HDDVs were equipped with selective catalytic reduction (SCR) systems and complied with the China V national emission standard. Two vehicles with lower cumulative mileage were numbered D1 and D2 (low-cumulative mileage vehicles, LMVs), while the other two with higher cumulative mileage were labeled D3 and D4

(high-cumulative mileage vehicles, HMVs). To assess the impact of ambient temperature on I/SVOC emissions, emission tests for D2 and D4 were conducted both at 0°C and 23°C. General information about the vehicles is presented in Table 1.

**Table 1. Information on the test fleet**

| Vehicle ID | D1 | D2 | D3 | D4 |
|---|---|---|---|---|
| Emission Standard | China V | China V | China V | China V |
| Aftertreatment Devices | SCR | SCR | SCR | SCR |
| Cumulative mileage ($\times 10^3$ km) | 22.21 | 34.84 | 169.50 | 188.33 |
| Gross Combined Weight Rating (GCWR, t) | 48.8 | 48.8 | 41.8 | 41.8 |
| Rated Power (kW) | 309 | 397 | 228 | 228 |
| Displacement (L) | 11.12 | 12.42 | 9.73 | 9.73 |

Each CHTC-TT lasts 1800 seconds, with an average speed of 46.6 km·h$^{-1}$ and a maximum speed of 88 km·h$^{-1}$ (Fig. S1). When the vehicles were driven at the speed specified by the CHTC-TT on the dynamometer, the emitted exhaust from tailpipes

would be diluted in the constant volume sampler (CVS). The diluted exhaust was detected for $CO_2$, CO, total hydrocarbons (THC), and $NO_x$ by the real-time gas analyzer module (MEXA-7400HLE, HORIBA, Japan) provided by the CATARC, and a series of offline sampling test samples were also collected from the CVS.

### 2.2 Sampling and analysis

The diluted exhaust from CVS was filtered with a 47 mm PTFE filter (R2PJ047, PALL Corporation, USA) and then collected



by the Tenax TA tubes (C1-AAXX-5003, MARKES International, UK) and 47mm quartz filters (Grade QM-A, Whatman, UK), respectively, for analyzing I/SVOCs and gas-phase organic compounds adsorbed on quartz filters ($Q_{gas}$, Fig. S2.). Notably, 2 TA tubes were connected in series for each sampling to prevent penetration, and the quantitative results of the two connected tubes were ultimately added together. Meanwhile, the particulate matters in the exhaust from CVS were also captured by another parallel pipe with a 47mm quartz filter ($Q_{total}$) for analyzing the particulate organic compounds, mainly including

I/SVOCs. However, since the quartz filters can adsorb gaseous organic compounds, the more accurate mass of particulate organics should be obtained by subtracting $Q_{gas}$ from $Q_{total}$. As shown in Sect. 3.2, $Q_{gas}$ accounted for 32% of $Q_{total}$. Thus, the total I/SVOC results in this paper were gaseous I/SVOCs collected by TA tubes plus particulate I/SVOCs collected by quartz filters after deducting artifacts (total I/SVOCs = TA + (1 - 32%) × $Q_{total}$). The TA tubes were prebaked at 320℃ for 2 hours and at 335℃ for 30 minutes with reverse nitrogen blowing in TC-20 (MARKES International, UK) to eliminate the risk of

contamination before sampling. The quartz filters were also prebaked for 8 hours at 550℃ in a muffle furnace to remove any carbonaceous contamination before sampling. All quartz filter samples have been stored in a constant temperature (23 ± 1 ℃) and humidity (50 ± 5 %) box until constant weight and then stored at -20℃.

    Each TA tube was injected with 2 μL of deuterated internal standard mixing solution (IS) through a mild nitrogen blow (CSLR, MARKES International, UK) before being analyzed by TD-GC×GC-MS. The TD-GC×GC-MS system was composed

of an autosampler with a thermal desorber (ULTRA-xr[TM] and UNITY-xr[TM], MARKES International, UK) and a solid-state modulator (SSM1810, J&X Technologies, China) installed on a gas chromatograph (8890, Agilent Technologies, USA) coupled with a mass spectrometer (5977 B, Agilent Technologies, USA). The quartz filter samples were also injected with the same IS and then put into clean empty glass tubes (C0-FXXX-0000, MARKES International, UK) to analyze by the TD-GC×GC-MS system.

In thermal desorber, TA tube (quartz filter) samples were heated at 320℃ (330℃) for 20 min with a trap flow of 50 mL·min$^{-1}$, and then all organics desorbed by heat were captured by the cold trap (U-T1HBL-2S, MARKES International, UK) with the carrier gas. Subsequently, the cold trap was heated at 330℃ (340℃) for 5 min so that the organics could enter GC×GC to separate and be detected by MS. In GC×GC, 4 different columns (Agilent Technologies, USA) were connected in series, from front to back: 30 m DB-5MS, 0.6 m VF-1ms, 0.7 m CP802510 (open tubular column), and 1.2 m DB-17MS. Among them,

VF-1ms switched between the cold and hot zones of the modulator. Hence, the organics that had undergone the first separation entered the subsequent columns in the form of a pulse for the second separation. The oven of 8890 and hot zone of the modulator matched the same heating program: maintained at 50℃ for 3 min, then increased to 310℃ at a rate of 5℃·min$^{-1}$ and maintained for 5 min. The cold zone of the modulator dropped from 9℃ to -51℃ at the fastest speed and maintained for 21.8 min, and then rose to 9℃ at a rate of 20℃·min$^{-1}$ and maintained for 34 min.

**2.3 Qualitative analysis and quantification of I/SVOCs**

    The I/SVOCs captured by TA tubes and quartz filters were identified and quantified with their respective authentic standards or surrogates using the three-step approach proposed by He et al. (2022b) Briefly, with more than fifteen hundred peaks of one chromatography scanned by GC×GC-MS, it was almost impossible to accurately identify and quantify every peak in the chromatography of each sample. Therefore, the organic compounds in the samples were categorized into eleven groups, each

with its specific mass spectrometry rule under electron energy 70 eV. According to these rules and retention time, peaks could be selectively screened from the total ion chromatogram (TIC). The peaks without external standard curves (ES) were quantified by the closest and same group ES for substitution. One hundred and twenty ES have been used in this study to cover as many organic compounds as possible, as shown in Fig. S3. All group identification codes and information on ES are listed in Table S1. Notably, the elution peak area that cannot be recognized by any identification code accounts for about 20% of the



total peak area, which has not been quantified in this study.

### 2.4 Emission factor calculation

All pollutant data were reported as distance-based and fuel-based emission factors (EFs):

$$EF_{d,i} = \frac{\Delta m_i}{s} \tag{1}$$

$$EF_{f,i} = \frac{\Delta m_i \cdot w_C}{12/44 \cdot \Delta CO_2 + 12/28 \cdot \Delta CO} \tag{2}$$

where $\Delta m_i$ was the measured background-corrected mass of species $i$ (mg). $s$ was the distance traveled by the vehicle in a

test cycle (km). $w_C$ was the measured carbon mass fraction of fuel, of 0.82. $\Delta CO_2$, $\Delta CO$ and were the background-corrected

masses of $CO_2$ and CO.

### 2.5 Modified combustion efficiency (MCE) calculation

MCE was applied herein to represent the combustion efficiency in each measurement, as displayed in:

$$MCE = \frac{\sum_{i=1}^{n} \frac{[CO_2]_i}{[CO_2]_i + [CO]_i}}{n} \tag{3}$$

where $[CO_2]_i$ and $[CO]_i$ are instantaneous mixing ratios of $CO_2$ and CO at second $i$, respectively, during the entire

cycle where $n$ is equal to 1800 s.2.6 Emission Inventory Calculation

According to the official guide (Ministry of Ecology and Environment of the People's Republic of China, 2024) and the

national HDDV population in 2022, the emission inventory was established based on:

$$E_n = \sum P \times VKT \times EF_n \tag{4}$$

where $E_n$ was the total mass of I/SVOC emissions of different cases in this study. $P$ was the vehicle population. $VKT$

represented the calibrated annual kilometers traveled per vehicle, which was considered 87786.15 km (240.51 km·d$^{-1}$ × 365 d)

for each freight vehicle (Anon, 2022). $EF_n$ was emission factor of I/SVOCs of different cases in mg·km$^{-1}$. 3 cases were

assumed in this study.

### 2.6 SOAFP estimation

The SOAFP derived from I/SVOCs was estimated followed the approach of Zhao et al (Zhao et al., 2015), and the detailed

parameterizations were listed in SI. The SOAFP (mg·km$^{-1}$) produced over a period ($\Delta t$) was calculated as follows:

$$SOAFP = \sum [EF_i \times (1 - exp(-K_{OH_i} \times [OH] \times \Delta t) \times Y_i] \tag{5}$$

where $EF_i$ was emission factor of pollution $i$ in mg·km$^{-1}$. $K_{OH_i}$ is the hydroxyl (OH) radical reaction rate constant of

compound $i$ at 25℃. $[OH]$ is the OH concentration, assumed to be $1.5 \times 10^6$ molecules·cm$^{-3}$. $\Delta t$ is the photooxidation time

(s). $Y_i$ is the SOA yield of precursors $i$.

## 3. Results and discussion

### 3.1 Overall results

The HDDV I/SVOC EFs ranged from 9 to 406 mg·km$^{-1}$ (41 to 1848 mg·kg-fuel$^{-1}$) in this study, consistent with previous

findings, indicating a broad range of I/SVOC EFs from HDDVs. For example, Zhao et al. (2015) reported the IVOC EFs of

assorted heavy-duty vehicles were 17 to 5354 mg·kg-fuel$^{-1}$, with various driving cycles and after-treatments. Similarly, He et

al. (2022b) manifested that the I/SVOC EFs of China IV and China VI HDDVs ranged from 38 to 18900 mg·kg-fuel$^{-1}$,

attributing this extensive range to the significant differences in after-treatments and emission standards of vehicles. Zhang et

al. (2024a) tested two China V HDDVs and reported that the gaseous I/SVOC EFs were 2034 and 2054 mg·kg-fuel$^{-1}$,

respectively.




To further analyze the I/SVOCs component and volatility distribution, the average EFs of all test cycles were divided into seven intervals based on $\log_{10}C^*$, as shown in Figure 1. Overall, IVOCs dominated the I/SVOC emissions with an average contribution of 81%, with the remaining 19% attributed to SVOCs. The primary contributors to the total identified EFs, ranked from high to low, were alkanes (including *n*- and *i*-alkanes, 20%), oxy-PAH & oxy-benzene (20%), phenol (14%), acid (11%), PAH_3rings (11%), alcohol (10%), and carbonyls (7%). The proportion of O-I/SVOC (including alcohols, phenols, carbonyls, acids, oxy-PAHs, and oxy-benzenes) accounted for 61% of the total. The proportions of other categories were lower than 5%. The alkane proportion was lower but O-I/SVOC proportion was higher than that in previous studies (alkane: 37% to 66%, O-I/SVOCs: 20% to 27%) (He et al., 2022b; Zhang et al., 2024a). The discrepancy may relate to variations in the tested vehicle types, as well as differences in the composition of diesel or lubricating oil used in this study. Most of the detected alkanes in this study were present in relatively higher-volatility *bins* like *bin* 6 ($\log_{10}C^* = 6$), while PAHs were distributed across *bin* 1 to 4. For O-I/SVOCs, alcohols and phenols mainly fell into *bin* 5, while oxy-PAHs & oxy-benzenes exhibited decreasing concentrations with decreasing volatility. Although a higher acid proportion was detected in this study than in previous studies (He et al., 2022b; Zhang et al., 2024a), their contribution to SOA production was considered minimal due to their low SOA yields (Huang et al., 2024).

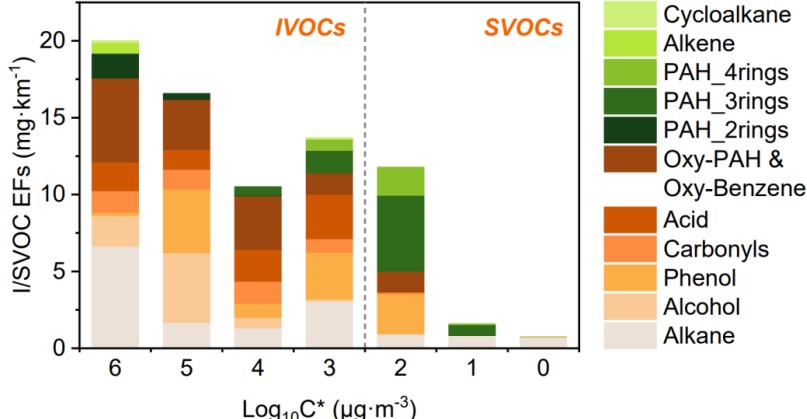

**Figure 1 Average volatility distribution of I/SVOCs from the tested fleet.** Different colored bars represent different organic groups.

### 3.2 Gas-particle partition and I/SVOC artifacts on quartz filters

Generally, the gaseous I/SVOCs consistently accounted for 79% to 99% of the total I/SVOCs, while particulate I/SVOCs contributed 1% to 21%. However, Liu et al. (2021) reported that China V HDDVs could emit more particulate I/SVOCs than gaseous I/SVOCs. This discrepancy may be attributed to the use of series sampling with a quartz filter and a TA tube, which can lead to the adsorption of a substantial fraction of gaseous I/SVOCs onto the quartz filters (artifacts), causing these compounds to be mistakenly categorized as particulate-phase. At the same time, the adsorption on the front quartz filters reduces the amount of gaseous I/SVOCs that reach the rear TA tubes, resulting in the final calculated proportion of particulate I/SVOCs exceeding 50%.

To assess the extent of adsorption artifacts on quartz filters, $Q_{gas}$ samples from the hot-start cycles of the tested vehicles were analyzed. Results indicated that artifacts accounted for 32% ± 14% of the mass fractions on quartz filters, aligning with findings from previous studies (May et al. 2013). Including these artifacts directly in the particulate-phase measurement introduces significant uncertainty into the calculated emission inventory, especially for IVOCs, and amplifies the uncertainties in



environmental impact predictions related to emission sources. As illustrated in Fig. S4(a), IVOCs dominated the artifacts on quartz filters, representing 98% of the mass, while SVOCs made up only 2%. From a chemical composition perspective, carbonyls were the most affected by adsorption artifacts (Fig. S4(b)). However, it remains challenging to determine the exact capacity of quartz filters to adsorb gaseous I/SVOCs or predict when saturation occurs, given the variability in filter properties across different manufacturers and production batches (Kirchstetter et al. 2001). Therefore, it is essential to minimize or eliminate particle quantification errors caused by adsorption artifacts to reduce uncertainties in subsequent modeling efforts. This is particularly crucial when assessing the environmental impact of IVOCs, given their substantial role in SOA formation and pollution forecasting.

**3.3 I/SVOC EFs and composition from HDDVs with varying cumulative mileage**

Figure 2(a) presents the distance-based EFs of I/SVOCs for HMVs and LMVs. The data reveal that the average I/SVOCs EFs of HMVs (D3&D4, 190 ± 94 mg·km$^{-1}$) were approximately eight times higher than those of LMVs (D1&D2, 23 ± 11 mg·km$^{-1}$), even HMVs consumed less fuel on average (26 L·100km$^{-1}$) compared to LMVs (33 L·100km$^{-1}$) for their lower GCWR. The significant disparity in I/SVOC EFs between HMVs and LMVs highlights cumulative mileage as a critical factor influencing I/SVOC EFs (p = 0.005 for hot-start cycles), which has often been overlooked in previous studies. For instance, the official guideline (Ministry of Ecology and Environment of the People's Republic of China, 2024) and COPERT 4 model, the latest vehicular emission factor model (Cai and Xie, 2013), do not account for the deterioration of organic emissions (e.g. THC) from diesel vehicles. In order to investigate the underlying causes of high I/SVOC emissions from HMVs, we compared the MCE of each test cycle. As shown in Fig. S5, a strong correlation (R² = 0.73) was observed between MCE and I/SVOC EFs. As combustion efficiency decreases, I/SVOC EFs rise, and HMVs exhibit greater variability in MCE than LMVs. This suggests that cumulative mileage contributes to increasing emissions and should be factored into emission inventories and SOA estimation. Given the scarcity of I/SVOC EF data in previous studies (Huang et al. 2013, Yao et al. 2015, Lv et al. 2020), we estimated the emission deterioration factors of I/SVOCs by leveraging the strong correlation between THC and I/SVOC emissions and available THC EFs. Figure 2(c) demonstrates a linear relationship (R2 = 0.9) between equivalent I/SVOCs and cumulative mileage. However, it is important to note that existing research primarily focuses on diesel vehicles with cumulative mileage below 200,000 km. Further experiments are necessary to determine whether I/SVOC emissions from HDDVs with over 200,000 km of mileage continue to increase linearly or if the trend stabilizes.

Additionally, we examined the cold-start extra emissions (CSEE), which was the difference between emissions from the cold-start cycle and hot-start cycle results. For HMVs, CSEE ranged from 657 to 5592 mg, whereas for LMVs, it ranged from 79 to 281 mg. CSEE contributed 18% to 59% of the total cold-start cycle emissions for HMVs and 21% to 45% for LMVs, respectively. It indicated that the I/SVOC emission deterioration could occur under both the cold-start and hot-running conditions.

To further analyze volatility and category distribution, the average EFs of HMVs and LMVs were plotted separately, as shown in Figure 3. The EF ratios across different volatility *bins* decreased with decreasing volatility, highlighting that the elevated I/SVOC EFs of HMVs were primarily due to a marked increase in organics within the volatility range of *bins* 2 to 6. Figure 3 further depicts the relative proportion of distinct organic groups present in I/SVOC emissions. O-I/SVOCs contributed 65% of the I/SVOCs emissions from HMVs, compared to 42% for LMVs. Since the SOA yields of O-I/SVOCs are lower than those of hydrocarbon-like I/SVOCs in the same *bin* (Chacon-Madrid and Donahue, 2011), variations in O-I/SVOC proportions directly impacted the SOAFP gap between HMVs and LMVs, which would be further discussed in Sect. 3.5. Alkane and oxy-PAH & oxy-benzene were the dominant contributors to I/SVOCs for both HMVs and LMVs. PAH_3rings contributed 8% of the I/SVOC emissions for HMVs, but 23% for LMVs. Interestingly, phenol, alkene, and cycloalkane were not detected in any





LMV samples.

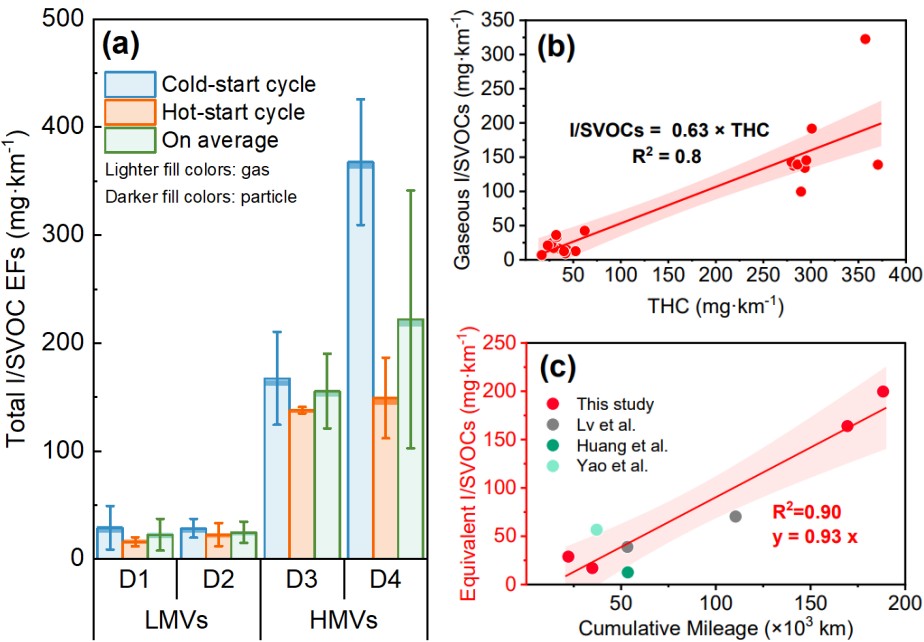

**Figure 2. (a) The bar chart represents total I/SVOC EFs from each HDDV under cold- and hot-start driving cycles.** The error bars are standard deviations. Gaseous and particulate I/SVOCs were represented by the lighter and darker fill colors respectively. The horizon axis is vehicle ID. **(b) The linear correlation between gaseous I/SVOC and THC EFs. (c)**

**The linear correlation between THC EFs and HDDV cumulative mileage.** Data are from this study and previous studies (Huang et al., 2013; Lv et al., 2020; Yao et al., 2015), of which tested vehicles shared the same THC emission limit (China IV/V and Euro IV/V emission standards limit diesel engines THC EF to 460 mg·kWh$^{-1}$) and similar weight.

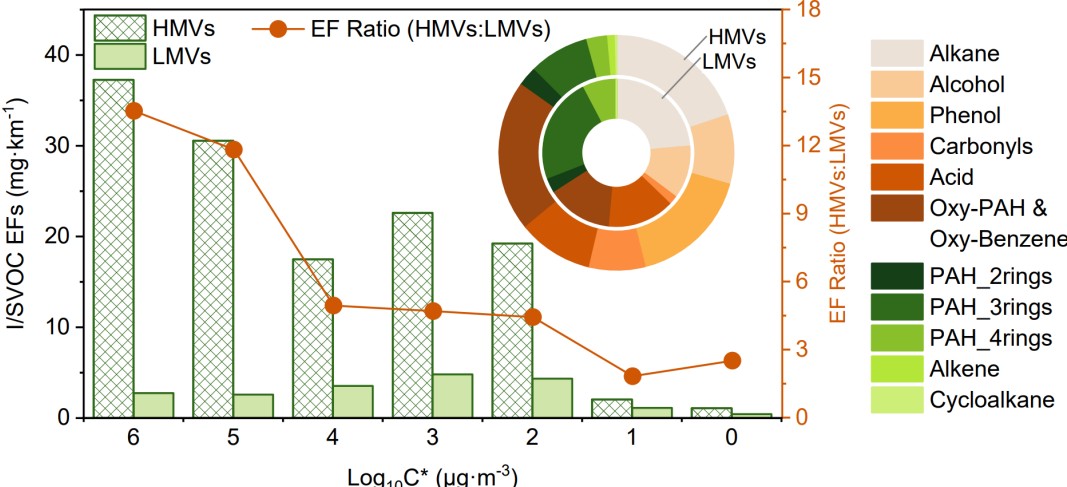

**Figure 3. Average volatility distribution of I/SVOCs of LMVs and HMVs.** The red dots represent the EF ratio of HMV

and LMV I/SVOCs (HMVs: LMVs). For the circular graph, different colored blocks represent the proportion of different organic groups in I/SVOCs, where the inner ring represents average data from LMVs and the outer ring from HMVs.





### 3.4 Low ambient temperature effect on total I/SVOC EFs and composition

Hot-start cycle I/SVOC emissions from LMV (D2) and HMV (D4) were tested at ambient temperatures of 0℃ and 23℃. As shown in Figure 4(a), colder ambient temperature increased the total I/SVOC EF of HMV from 127 mg·km$^{-1}$ to 171 mg·km$^{-1}$

(p = 0.01). In contrast, no statistically significant increase was observed for the LMV (p = 0.23). Figure S6 shows the strong linear correlation (R$^2$ = 0.93) between I/SVOC EFs and MCE for LMVs and HMVs across different ambient temperatures. This finding further supports that the decline in MCE may be a direct cause of the increase in I/SVOC EFs. Additionally, the MCE of LMVs demonstrated greater stability, which explains the absence of elevated I/SVOC emissions at low ambient temperatures in comparison to HMVs. This suggests that prolonged engine use enhances the sensitivity of I/SVOC emissions

to ambient temperature. Even in the absence of instantaneous emission data of I/SVOCs at different ambient temperatures, the strong linear correlation between THC and I/SVOCs allows us to infer the instantaneous THC emission profile. Figure 4(b) illustrates that HMV was likely to emit higher I/SVOC levels than LMV during rapid acceleration phases at 0℃, such as those occurring from 210 s to 220 s or from 1011 s to 1032 s along the speed curve. Furthermore, prior study has demonstrated that low temperatures significantly affect VOC emissions from diesel vehicles during cold-start conditions (Dardiotis et al., 2013).

Therefore, we recommend that future research should focus on the I/SVOC emissions of vehicles during low-temperature cold-start.

Regarding the distribution of I/SVOC categories, the mass fraction of PAHs increased at lower temperatures for both vehicle types (LMV: from 17% to 52%, HMV: from 10% to 14%). Given the toxicity of PAH, further research on the changes in exhaust gas toxicity in low-temperature environments is warranted, as the elevated PAH emissions may result from incomplete

combustion under cold conditions. Additionally, the proportion of O-I/SVOCs in HMV starkly increased from 52% to 78%, while no such trend was observed in LMV. Within the O-I/SVOCs of HMV, there was a notable decrease in alcohol, accompanied by a significant increase in carbonyls and oxy-PAH & oxy-benzene from 23℃ to 0℃. This substantial increment of O-I/SVOC is expected to influence the SOA yield, as O-I/SVOCs typically exhibit lower SOAFP compared to hydrocarbon-like organics, such as alkanes (Chacon-Madrid and Donahue, 2011).

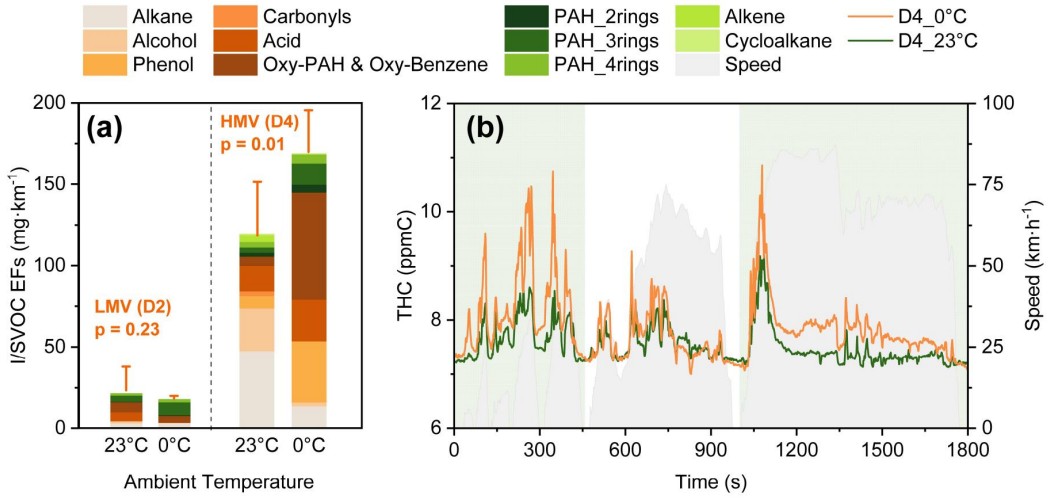


**Figure 4. (a) Total I/SVOC EFs of D2 and D4 at 0℃ and 23℃. Different colored bars represent different organic groups. (b) The average instantaneous THC emission concentration of D4 at 0℃ (orange line) and 23℃ (green line).**

### 3.5 SOAFP of the I/SVOCs

To evaluate the environmental impact of HDDV exhaust, Figure 5 depicts the average potential SOA production after 48



hours of photooxidation for the tested vehicles. After two days of photooxidation, the estimated SOAFP of HMVs reached 30 mg·km$^{-1}$, approximately four times higher than that of LMVs (8 mg·km$^{-1}$). However, the four-fold increase in SOAFP with cumulative mileage was less pronounced compared to the eight-fold increase observed for I/SVOC EFs. This discrepancy is primarily attributed to the greater increase in O-I/SVOC EFs relative to hydrocarbon-like organics such as alkanes, despite the lower SOA yields of O-I/SVOCs (Chacon-Madrid and Donahue, 2011) (Sect. 3.3, Figure 3(b)). The largest contributors to

SOAFP for HMVs were alkane (19%), oxy-PAH & oxy-benzene (18%), and phenol (18%), whereas, for LMVs, they were alkane (26%), acid (17%), and PAH_3rings (17%). Therefore, alkane, oxy-PAH & oxy-benzene, and phenol can be identified as the key contributors driving the increase in SOAFP for HMVs.

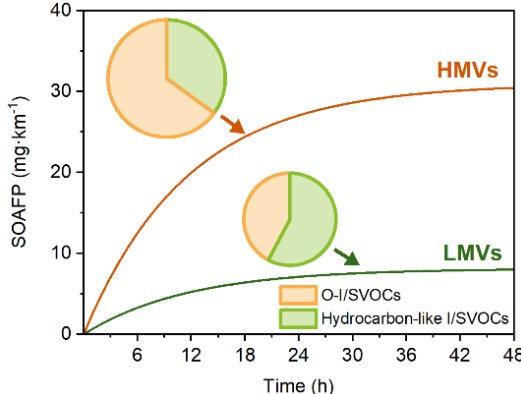

**Figure 5. The average SOAFP for HMVs and LMVs during 48 h photooxidation.** The pie charts represent the
295              contribution of hydrocarbon-like I/SVOCs and O-I/SVOCs to the total I/SVOC emissions of HMVs and LMVs.

To estimate the effects of cumulative mileage and ambient temperature on the I/SVOC emission inventory, we constructed an emission inventory of China V HDDVs as a case study. In scene 1, we utilized the average I/SVOC EFs of all tested vehicles, consistent with the approach taken in previous studies (Liu et al., 2021; Zhao et al., 2022; Wu et al., 2019; Zhang et al., 2024b); in scene 2, the calculation was based on the assumption that the EFs continuously increase linearly with cumulative mileage

(Sect. 3.3); scene 3 expanded on scene 2, assuming an average temperature of 0°C for three months within a year, thereby accounting for the low temperatures on I/SVOC emissions.

In 2022, the estimated I/SVOC emissions from China V HDDVs were 20, 60, and 66 kt for scenes 1, 2, and 3, respectively. The emissions in scene 2 were up to three times higher than that in scene 1. When considering the impact of low temperatures as in scene 3, the total emissions increased by an additional 10%. Given the critical role of accurate HDDV I/SVOC emission

inventories in predicting urban SOA formation, it is recommended that future studies measure and track I/SVOC emissions from HDDVs over a longer term (exceeding 3 years or corresponding to higher cumulative mileage) to better understand the true potential and patterns of I/SVOC emission degradation from diesel vehicles.

## 4 Conclusions

In this study, gaseous and particulate I/SVOCs emitted from four HDDVs were comprehensively analyzed using TD-
GC×GC-MS. The results indicated that the EFs of I/SVOC EFs from HDDVs ranged from 10 to 409 mg·km$^{-1}$, with gaseous I/SVOCs contributing between 79% to 99%, while particulate I/SVOCs contributing 1% to 21%. A key finding was the significant impact of vehicle cumulative mileage on I/SVOC emissions, with HMVs emitting eight times more I/SVOCs than



LMVs. This suggests that emission deterioration factors should be incorporated into emission inventories for more accurate predictions of SOA formation. A linear relationship between I/SVOC emissions and vehicle cumulative mileage was also established, emphasizing the need for long-term emission monitoring of HDDVs. Deterioration of I/SVOC emissions could occur under both cold-start and hot-running conditions, with comparable proportions of I/SVOC emissions during the cold-start cycles of HMVs and LMVs. Furthermore, volatility and category distribution analysis revealed that the increase in I/SVOC emissions from HMV was primarily driven by higher-volatility compounds (*bins* 2 to 6). Phenol was found to be the third most abundant in HMV emissions, whereas phenol, alkene, and cycloalkane were also not detected in LMV emissions.

Low ambient temperatures increased I/SVOC emissions from HMVs but not from LMVs, likely due to the prolonged use of engines. A strong linear correlation ($R^2$ = 0.93) between I/SVOC EFs and MCE from LMVs and HMVs across various temperatures suggests that the decline in combustion efficiency may be a direct cause of the increase in I/SVOC EFs. Changes in the composition of I/SVOCs at low temperatures were observed, particularly an increase in PAHs and oxygenated compounds, both of which can influence SOA formation.

Finally, the SOAFP estimations revealed that the SOAFP of HMVs was approximately four times more than that of LMVs after 48 hours of photooxidation. Additionally, a China V I/SVOC emission inventory was established based on various assumptions. Results indicated that neglecting emission discrepancy between LMVs and HMVs could lead to a threefold underestimation of inventory, and accounting for low temperature would further increase the total emissions by an additional 10%. The study recommends incorporating cumulative mileage and temperature effects into future emission inventories for more accurate predictions of urban SOA formation.

**Associated content**

**Supporting information**

Additional experimental details, description of sampling sites, supplementary results, and supporting tables and figures.

**Author Contributions**

S.G.: Experiment, formal analysis, data validation, writing−original draft; X.Z.: Writing−reviewing and editing, project administration, supervision, funding acquisition; X.H.: Model development and funding acquisition; L.Z.: Experiment and funding acquisition; L.H. and X.W.: Experiment; Y.D.: Data validation; Z.H., T.C., and S.X.: Experiment; Y.Y.: Funding acquisition; S.X.: Editing; S.Z., J.J., and Y.W.: Data validation, writing−reviewing and editing.

**Notes**

The authors declare no competing financial interests.

**Acknowledgments**

The authors acknowledge the financial support of the National Natural Science Foundation of China (grant nos.51978404, 42105100, 42307136, and 42261160645), Macao Science and Technology Development Fund (0023/2022/AFJ and 001/2022/NIF), and the Scientific Research Fund at Shenzhen University (grant nos. 868-000001032089 and 827-000907).

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
