# Peer review of "Emissions of Intermediate- and Semi-Volatile Organic Compounds (I/SVOCs) from Different Cumulative Mileage Diesel Vehicles under Various Ambient Temperatures"

_EGUsphere, 2024_

## Author Comment (AC1)

Point-to–point responses to review comments (egusphere-2024-3290)

Title: Emissions of Intermediate- and Semi-Volatile Organic Compounds (I/SVOCs) from Different
Cumulative Mileage Diesel Vehicles under Various Ambient Temperatures

Review of Shuwen Guo et al.

In the current study, gaseous and particulate I/SVOCs emitted from four HDDVs were analyzed
using GC×GC-MS. The emission factors as well as the composition of I/SVOCs were reported.
Overall, the experiments were nicely done and the data are well analyzed. The current contribution
is a welcome addition to the field. There are several places in the paper are a bit obscure as detailed
in the comments below. Beyond these, I do not see any major obstacles to publication.

Response:

We thank the reviewer for supporting our work, and we provide below a point-by-point response
to the individual comments.

Specific comments:

(1) The experiments were well organized. My general question is the innovation of the current study.
The I/SVOC emissions as well as their compositions from heavy diesel vehicles have been widely
reported, including the studies from their own group. Any new findings that the authors would like
to highlight in the current one?

Response:

Thanks for the comment. We highlight the findings below:

1) This study discussed the differences in I/SVOC emissions of heavy-duty diesel vehicles
(HDDVs) with different cumulative mileage and calculated the emission deterioration
coefficient, which has been overlooked in previous studies (Chang et al., 2022; He et al., 2022a,
b; Liu et al., 2021). We found that overlooking the I/SVOC emission degradation will result in
a more than three-fold underestimation of the total I/SVOC emissions of China V HDDVs in
China.
2) The low ambient temperatures would lead to more I/SVOC emissions for high-mileage vehicles
(HMVs), but no significant impact on low-mileage vehicles (LMVs). This has also been less
focused on in the past.
3) We discussed the certain linear correlation ($R^2$ = 0.73) between I/SVOC EFs and modified
combustion efficiency (MCE), which reveals the increase of I/SVOC emissions above is caused
by a decrease in engine combustion efficiency.

(2) Line 24: "Compounds such as phenol, …. appeared only in HMV emissions". These are
compounds that are widely observed in vehicle emissions. Any reason for their disappearance in
LMV emission? Are there any potential artifacts in the sample analysis?

Response:

Thanks for the comment. We did observe chromatographic peaks of phenols from LMV exhaust,
but their peak areas were significantly smaller compared to those observed in HMVs and were indistinguishable from those in background samples. Phenols reported in previous studies were also
emitted by high cumulative mileage vehicles. For instance, He et al. (2022) investigated HDDVs
with a service duration ranging from 2 to 6 years. Zhang et al. (2024) tested I/SVOCs from HDDVs
with service duration ranging from 1 to 6 years and found weak phenol signals for newer ones. As
vehicle age and mileage increase, engine combustion efficiency decreases, leading to higher
organic emissions in I/SVOCs and thus phenol concentrations above background levels to be
detected.

(3) Line 179: "Oxy-PAH&Oxy-benzene", I don't think the abbreviations were pre-defined. Also,
what compounds specifically do they represent? I noticed the authors also separately classify
"phenol" instead of grouping them into "Oxy-benzene".

Response:

Thanks for the comment. Similar abbreviations have appeared in our previous research papers (He
et al., 2022a, b 2024), and we apologize for the author's oversight in not specifying the types of
organic compounds covered by the abbreviations again in the manuscript of this study. Oxy-PAH
& Oxy-benzene represent all organic compounds containing benzene rings and oxygen-containing
groups, except for phenols whose hydroxyl group is directly connected to the benzene ring. We
supplement a detailed description of all organic category names as SI-1 in supporting information
(SI).

**"SI-1. Description of all organic category names.**

Alkane: n-alkane and i-alkane. Alcohol: aliphatic alcohol. Phenol: organics containing one benzene
ring and a hydrocarbon group directly attached to the benzene ring. Carbonyls: aliphatic ketone and
aliphatic aldehyde. Acid: aliphatic acid. Oxy-PAH & Oxy-benzene: organic compounds containing
benzene rings and oxygen-containing groups, except for phenols whose hydroxyl group is directly
connected to one benzene ring. PAH_2rings: PAH with 2 benzene rings. PAH_3rings: PAH with 3
benzene rings. PAH_4rings: PAH with 4 benzene rings. Alkene: organics containing carbon double
bond(s) without any other function groups. Cycloalkane: organics containing a saturated carbon
ring without any other function groups.

"

Please refer to lines 27-35 in the SI for details.

(4) Line 190: the emission factors of ISVOCs between LMV and HMV differs quite a lot according
to Figure 2a. And according to Figure 3, the fractional contributions from different components are
also different for HMV and LMV. Hence, I'm not sure it is appropriate to present the average
volatility distributions of I/SVOCs from the entire fleet. Could Figure 1 be separated into LMV and
HMV?

Response:

Suggestion taken. We divided Figure 1 into LMV and HMV results for plotting, rather than taking
the average of the two, as shown in the figure below. Figure (a) was the average volatility
distribution of I/SVOCs from the LMVs, and (b) from HMVs. Different colored bars represent different organic groups. The highest I/SVOC EFs for LMVs were in bin 3 and bin 2, reaching 5
mg·km$^{-1}$ and 4 mg·km$^{-1}$; but for HMVs were in bin 6 and bin 5, reaching 37 mg·km$^{-1}$ and 31 mg·km$^{-1}$
$^{1}$, respectively. The I/SVOCs they emitted are mainly IVOCs.

The information in this modified figure overlapped with Figure 3 in the main context (line 253),
and thus this modified figure was placed in the SI as Fig. S4. The original Fig. 1 in line 190 of the
manuscript has been deleted, and other figure numbers and figure references in the main text have
been modified. The SI was modified as follows:

"

[Figure]

Fig. S4. The average volatility distribution of I/SVOCs from the (a) LMVs and (b) HMVs.
    Different colored bars represent different organic groups."

Please refer to lines 76-79 in the SI for details.

(5) Line 225: How many sets of the tests were performed? Does each data point on Figure 2b
represents the average emission factor for each entire 1800s test cycle? Also, I hesitate to agree that
gaseous I/SVOCs show good correlation with THC because the datapoints on Figure 2b
concentrates at two ends of the fitted line, which might affect the reliability of the linear regression.
Any more evidence on this point? Or any other supporting references?

Response:

Thanks for the comment. We performed ten sets of tests and two or three parallel tests were
conducted, as shown in the table below, which has been supplemented in the SI. Each data point on
Figure 2b represents the EF for each entire 1800s test cycle.

Previous studies on vehicle exhaust emissions have reported a linear correlation between I/SVOC
and THC (or nonmethane hydrocarbons, NMHCs). For example, Zhao et al. (2015, 2016) reported
a stronger correlation between total I/SVOCs and NMHCs (R$^2$ = 0.92-0.98) emitted from motor
vehicles. The strong correlation between total I/SVOCs and THC (R$^2$ = 0.78-0.87) was also found
by Tang et al. (2021). Their linear correlation and ratio have also been used to estimate the I/SVOC
emission inventory when no detailed I/SVOC measurement data are available (Zhang et al., 2024;
Zhao et al., 2022). For instance, Zhao et al. (2022) used the EF ratios of IVOCs to NMHCs of diesel vehicles calculated by previous studies to estimate the IVOC emission inventory for mobile sources
in China.

"Table S1. Sets of test cycles.

| NO. | Vehicle ID | Ambient Temperature | Cold- or Hot-start Cycle | Repetitions |
|---|---|---|---|---|
| 1 | D1 | 23°C | Cold-start cycle | 2 |
| 2 | D1 | 23°C | Hot-start cycle | 2 |
| 3 | D2 | 23°C | Cold-start cycle | 2 |
| 4 | D2 | 23°C | Hot-start cycle | 2 |
| 5 | D2 | 0°C | Hot-start cycle | 2 |
| 6 | D3 | 23°C | Cold-start cycle | 3 |
| 7 | D3 | 23°C | Hot-start cycle | 3 |
| 8 | D4 | 23°C | Cold-start cycle | 3 |
| 9 | D4 | 23°C | Hot-start cycle | 3 |
| 10 | D4 | 0°C | Hot-start cycle | 2 |

Please refer to lines 61 in the SI for details.

(6) The influence of temperature on emission is interesting. What are the variations of other
pollutants with the changes in temperature, i.e., THC, NOx, CO, etc?

Response:

We did find that the emissions of other pollutants from HMV were also affected by low ambient
temperature, but this part was not mentioned in the manuscript as it is not related to the title of the
article. The low temperature caused the average EFs of THC (NOx, CO) to increase from 289
mg·km$^{-1}$ (2629 mg·km$^{-1}$, 359 mg·km$^{-1}$) to 302 mg·km$^{-1}$ (3555 mg·km$^{-1}$, 404 mg·km$^{-1}$). However, the
reasons for their increase in EFs were different. THC and CO are the by-products of incomplete
combustion of diesel, and thus their EFs are directly related to MCE. The EF of NOx is related to
the treatment efficiency of the selective catalytic reduction (SCR) system, whose ammonia aqueous
solution may partially solidify at 0°C thereby reducing the reaction efficiency of NOx in SCR. The
same emission change pattern of these conventional pollutants was found in LMV. The detailed
data is shown in the table below, which has been supplemented in SI.

"Table S3. Average THC, NOx, and CO EFs for LMV and HMV, respectively.

| Vehicle | Test Cycle | THC (mg·km$^{-1}$) | NOx (mg·km$^{-1}$) | CO (mg·km$^{-1}$) |
|---|---|---|---|---|
| LMV (D2) | Hot_23°C | 35 | 6951 | 600 |
| | Hot_0°C | 38 | 8048 | 657 |
| HMV (D4) | Hot_23°C | 289 | 2629 | 359 |
| | Hot_0°C | 302 | 3555 | 404 |

"

Please refer to lines 65 in the SI for details.

(7) The overall presentation is acceptable, but English could do with improvement in places.

Response:

Thanks for the comment. We have polished the English expression of the entire text again. Taking
the modifications listed in the table below for example:

| Original | | Modified | |
|---|---|---|---|
| *Line* | *Text* | *Line* | *Text* |
| 49 | These variations underscore the need for a more precise assessment of diesel vehicle I/SVOC emission factors (EFs). | 47 | These discrepancies highlight the urgent need for a more precise assessment of diesel vehicle I/SVOC emission factors (EFs). |
| 52 | Furthermore, many regions in China experience temperatures of 0°C or lower during the autumn and winter. Consequently, HDDVs operating under such low-temperature conditions may exhibit different emission characteristics compared to those under normal temperatures (e.g., 23°C). This underscores the importance of examining the variations in I/SVOC emissions and exhaust component distribution from HDDVs across different temperature conditions. | 50 | Given that many regions in China experience temperatures below 0°C during winter, evaluating how HDDVs operate under such conditions is critical in I/SVOC emissions and exhaust component distribution across different temperature conditions. |
| 61 | In the study of Zhao et al (2015), approximately 80% of I/SVOCs emitted by diesel vehicles remain unresolved by GC-MS, reckoned as UCM. Moreover, due to the variability in the response signals detected by mass spectrometry for different complex organic compounds (He et al., 2022b), the lack of detailed component information introduces significant uncertainties in I/SVOC quantification and prediction of SOA formation potential (SOAFP). | 58 | For example, Zhao et al (2015) reported that 80% of I/SVOCs emitted by diesel vehicles were classified as UCM. This lack of detailed chemical information introduces uncertainties in I/SVOC quantification and prediction of SOA formation potential (SOAFP) (He et al., 2022b). |
| 182 | The alkane proportion was lower but O-I/SVOC proportion was higher than that in previous studies (alkane: 37% to 66%, O-I/SVOCs: 20% to 27%) (He et al., 2022b; Zhang et al., 2024a). | 180 | The proportion of O-I/SVOCs was notably higher in this study compared to previous research, where alkanes typically accounted for 37% to 66% and O-I/SVOCs for 20% to 27% (He et al., 2022b; Zhang et al., 2024a). |

*References:*

*Chang, X., Zhao, B., Zheng, H., Wang, S., Cai, S., Guo, F., Gui, P., Huang, G., Wu, D., Han, L.,*
*Xing, J., Man, H., Hu, R., Liang, C., Xu, Q., Qiu, X., Ding, D., Liu, K., Han, R., Robinson, A. L.,*
*and Donahue, N. M.: Full-volatility emission framework corrects missing and underestimated*
*secondary organic aerosol sources, One Earth, 5, 403–412,*
*https://doi.org/10.1016/j.oneear.2022.03.015, 2022.*

*He, X., Zheng, X., Zhang, S., Wang, X., Chen, T., Zhang, X., Huang, G., Cao, Y., He, L., Cao, X.,*
*Cheng, Y., Wang, S., and Wu, Y.: Comprehensive characterization of particulate intermediate-*
*volatility and semi-volatile organic compounds (I/SVOCs) from heavy-duty diesel vehicles using*
*two-dimensional gas chromatography time-of-flight mass spectrometry, Atmos. Chem. Phys., 22,*
*13935–13947, https://doi.org/10.5194/acp-22-13935-2022, 2022a.*

*He, X., Zheng, X., You, Y., Zhang, S., Zhao, B., Wang, X., Huang, G., Chen, T., Cao, Y., He, L.,*
*Chang, X., Wang, S., and Wu, Y.: Comprehensive chemical characterization of gaseous I/SVOC*
*emissions from heavy-duty diesel vehicles using two-dimensional gas chromatography time-of-*
*flight mass spectrometry, Environ. Pollut., 305, 119284,*
*https://doi.org/10.1016/j.envpol.2022.119284, 2022b.*

*He, X., Zheng, X., Guo, S., Zeng, L., Chen, T., Yang, B., Xiao, S., Wang, Q., Li, Z., You, Y., Zhang,*
*S., and Wu, Y.: Automated compound speciation, cluster analysis, and quantification of organic*
*vapors and aerosols using comprehensive two-dimensional gas chromatography and mass*
*spectrometry, Atmos. Chem. Phys., 24, 10655–10666, https://doi.org/10.5194/acp-24-10655-2024,*
*2024.*

*Liu, Y., Li, Y., Yuan, Z., Wang, H., Sha, Q., Lou, S., Liu, Y., Hao, Y., Duan, L., Ye, P., Zheng, J., Yuan,*
*B., and Shao, M.: Identification of two main origins of intermediate-volatility organic compound*
*emissions from vehicles in China through two-phase simultaneous characterization, Environmental*
*Pollution, 281, 117020, https://doi.org/10.1016/j.envpol.2021.117020, 2021.*

*Tang, R., Lu, Q., Guo, S., Wang, H., Song, K., Yu, Y., Tan, R., Liu, K., Shen, R., Chen, S., Zeng, L.,*
*Jorga, S. D., Zhang, Z., Zhang, W., Shuai, S., and Robinson, A. L.: Measurement report: Distinct*
*emissions and volatility distribution of intermediate-volatility organic compounds from on-road*
*Chinese gasoline vehicles: implication of high secondary organic aerosol formation potential,*
*Atmos. Chem. Phys., 21, 2569–2583, https://doi.org/10.5194/acp-21-2569-2021, 2021.*

*Zhang, Z., Man, H., Zhao, J., Huang, W., Huang, C., Jing, S., Luo, Z., Zhao, X., Chen, D., He, K.,*
*and Liu, H.: VOC and IVOC emission features and inventory of motorcycles in China, Journal of*
*Hazardous Materials, 469, 133928, https://doi.org/10.1016/j.jhazmat.2024.133928, 2024.*

*Zhao, Y., Nguyen, N. T., Presto, A. A., Hennigan, C. J., May, A. A., and Robinson, A. L.: Intermediate*
*volatility organic compound emissions from on-road diesel vehicles: chemical composition,*
*emission factors, and estimated secondary organic aerosol production, Environ. Sci. Technol., 49,*
*11516–11526, https://doi.org/10.1021/acs.est.5b02841, 2015.*

*Zhao, Y., Nguyen, N. T., Presto, A. A., Hennigan, C. J., May, A. A., and Robinson, A. L.: Intermediate*
*volatility organic compound emissions from on-road gasoline vehicles and small off-road gasoline*
*engines, Environ. Sci. Technol., 50, 4554–4563, https://doi.org/10.1021/acs.est.5b06247, 2016.*

*Zhao, J., Qi, L., Lv, Z., Wang, X., Deng, F., Zhang, Z., Luo, Z., Bie, P., He, K., and Liu, H.: An*
*updated comprehensive IVOC emission inventory for mobile sources in China, Science of The Total*
*Environment, 851, 158312, https://doi.org/10.1016/j.scitotenv.2022.158312, 2022.*

---

## Author Comment (AC2)

Point-to–point responses to review comments (egusphere-2024-3290)

Title: Emissions of Intermediate- and Semi-Volatile Organic Compounds (I/SVOCs) from Different
Cumulative Mileage Diesel Vehicles under Various Ambient Temperatures

Review of Shuwen Guo et al.

This manuscript investigates the gaseous and particulate I/SVOC emission factors (EFs) of high-
mileage vehicles (HMVs) and low-mileage vehicles (LMVs) under varying ambient temperatures
using TD-GC×GC-MS. The authors provide a comprehensive analysis of the variations in emission
factors and their chemical components, identifying a linear correlation between I/SVOC EFs and
the modified combustion efficiency (MCE). The experimental methodology is thorough and
reliable, and the findings present a significant advancement in understanding I/SVOC emissions
from heavy-duty diesel vehicles (HDDVs). The results have practical implications, particularly for
researchers developing I/SVOC emission inventories and secondary organic aerosol (SOA)
prediction. Overall, I recommend accepting this manuscript following minor revisions.

Response:

Sincerely thanks for the positive comments. We have carefully revised the manuscript according to
the specific comments.

Specific comments:

1. In section 2.1, the authors describe the information of the four in-use HDDVs. But the vehicle
brand and the engine model of the HDDVs, which are closely related to the vehicle emission
are not given. The related information should be given, and some discussion about the
uncertainty caused by these differences and the aging of the engine should be included in the
manuscript.

Response:

Thanks for the suggestion. We have modified Table 1 in the main text, which offers more
information including the brand, the engine model, and the in-use duration to represent the aging
of the engine. Also, the uncertainty caused by them has been discussed.

"**Table 1. Information on the test fleet**

| Vehicle ID | D1 | D2 | D3 | D4 |
|---|---|---|---|---|
| **Emission Standard** | China V | China V | China V | China V |
| **Aftertreatment Devices** | SCR | SCR | SCR | SCR |
| **Brand** | DONGFENG | SINOTRUK | DELONG | DELONG |
| **Engine Model** | dCi450-51 | MC13.54-50 | WP10.310E53 | WP10.310E53 |
| **In-use Duration** | 7 months | 8 months | 32 months | 32 months |
| **Cumulative Mileage ($\times 10^3$ km)** | 22.21 | 34.84 | 169.50 | 188.33 |

| Gross Combined Weight Rating (GCWR, t) | 48.8 | 48.8 | 41.8 | 41.8 |
|---|---|---|---|---|
| Rated Power (kW) | 309 | 397 | 228 | 228 |
| Displacement (L) | 11.12 | 12.42 | 9.73 | 9.73 |

…

…It should be noted that existing research primarily focuses on diesel vehicles with cumulative mileage below 200,000 km. Further experiments are necessary to determine whether I/SVOC emissions from designated HDDVs with over 200,000 km of mileage continue to increase linearly or stabilize. Also, the brand, engine models, GCWR, and displacement of the four HDDVs were slightly different (Table 1), which might bring some uncertainty to the emission analysis results (Zeng et al., 2024; Tolouei and Titheridge, 2009; Aosaf et al., 2022). Future studies should further consider the uncertainties brought by these factors."

Please refer to lines 88 and 226-229 in the main text for details.

2. In Section 2.1, it is better to give the dilution ratio of the exhaust. In addition, was the temperature of the sampling pipe maintained as a certain level to reduce thermophoretic and condensational losses?

Response:

Thanks for the comment. The dilution ratio of the exhaust was about 40 of the exhaust. CVS is a constant temperature dilution system that stabilizes the airflow within the sampling channel at 25°C to reduce thermophoretic and condensational losses.

"Each CHTC-TT lasts 1800 seconds, with an average speed of 46.6 km·h$^{-1}$ and a maximum speed of 88 km·h$^{-1}$ (Fig. S1). When the vehicles were driven at the speed specified by the CHTC-TT on the dynamometer, the emitted exhaust from tailpipes was diluted in the constant volume sampler (CVS). The exhaust dilution ratio was about 40. The CVS system maintains the airflow of the diluted exhausts at 25°C to avoid thermophoretic and condensational losses. $CO_2$, CO, total hydrocarbons (THC), and $NO_x$ from the diluted exhaust were detected by the real-time gas analyzer module (MEXA-7400HLE, HORIBA, Japan) provided by the CATARC, and a series of offline sampling test samples were also collected from the CVS."

Please refer to lines 91-92 in the main text for details.

3. In Section 2.2, the authors describe the method to remove effects of absorption on the quartz filter when calculating the total I/SVOCs. The gas phase of I/SVOCs are collected after a PTFE filter, but the separation of gas and particle I/SVOCs after the PTFE may break the equilibrium of gas/particle I/SVOCs and lead to evaporation of particle I/SVOCs, which may overestimate the Qgas. It is better to provide some discussion on the uncertainty of this method.

Response:

Suggestion taken. We added the following text in line 105-110 to address this issue in the revised manuscript:

"…the total I/SVOC results in this paper were gaseous I/SVOCs collected by TA tubes plus particulate I/SVOCs collected by quartz filters after deducting artifacts (total I/SVOCs = TA + (1 - 32%) × $Q_{total}$). Notably, the gas phase of I/SVOCs was collected after passing through a PTFE filter, and the separation of gas and particle I/SVOCs beyond the PTFE filter may disrupt the equilibrium between them. Cheng et al. (2010) evaluated the collection artifacts of organic carbon using various quartz filter sampling methods and found that about 10% of the Qgas derived from volatilized particulate organic carbon by the sampling method used in this study. Therefore, the Qgas in this study may be slightly overestimated. The TA tubes were prebaked at 320°C for 2 hours…"

4. The sentence "…by He et al. (2022b)" in line 132 is missing a period at the end. Please correct this.

Response:

Thanks for pointing this out. The error here has been corrected. Please refer to line 131 in the main text.

5. Some phrases should use standard abbreviations. For instance, the phrase "...as shown in Figure 1" in line 177 should be revised to "...as shown in Fig. 1." Ensure consistent abbreviation usage throughout the manuscript.

Response:

Thanks for pointing this out. Figure 1 in the original manuscript has been removed based on the opinion of Referee #1 and has been revised as Fig. S4 in SI. Therefore, the sentence has been revised to "… as shown in Fig. S4.". Please refer to line 175 in the main text. The abbreviation in the context has been rechecked.

6. In line 227, the "2" in "R2=0.9" is not in superscript. Please adjust the formatting for accuracy.

Response:

Thanks for pointing this out. The error here has been corrected. Please refer to line 223 in the modified manuscript.

7. The steps for qualitatively identifying organic compounds using mass spectrometry principles require further elaboration. Could the authors provide a detailed description of these procedures in the methods section?

Response:

Thanks for the comment. Taking alkanes as an example, compounds containing hydrocarbon chains give rise to a series of ions separated by 14 Da (-CH2-), as shown in Fig. 1. As a result, the top ions to identify alkanes would be m/z = 43, m/z = 57, m/z = 71, and m/z = 84. Due to the stability of chemical groups, generally, the abundance of m/z = 57 is highest, followed by m/z = 43 and m/z = 71. When incorporating these rules into the data treatment software (Canvas, version 2.5, J&X Technologies), a few steps need to be taken, as shown in Fig. 2. Four built-in features can be deployed. ABUND (X) returns the normalized abundance of the input ion mass; HASMASS (X)
returns the value to indicate if the input ion exists; ORDER (X) returns the order of the input ion
mass; MASS (X) returns the mass of the input ion's order. Additionally, the function allows two
logical operators, "And" and "Or". Then, the cluster of alkanes can be extracted by the following
rules:

((MASS(1)=43 && (MASS(2)=57 ‖ MASS(2)=71 ‖ MASS(2)=41)) ‖ (MASS(1)=57 &&
(MASS(2)=43 ‖ MASS(2)=71 ‖ MASS(2)=41)))

where "&&" and "‖" refer to the logical operators "And" and "Or", respectively. Paste the rules in
Ion Extractor Editor and the cluster of alkanes can be filtered, as shown in Fig. 3.

[Figure]

Figure 1. The common fragmentation patterns of n-alkanes.

[Figure]

Figure 2. The steps to enable the ion extract function built in Canvas.

[Figure]

Figure 3. Comprehensive two-dimensional chromatograms before and after screening alkanes. Each red and blue point represents a chromatographic peak.

We have compiled the above explanation in SI. Please refer to line 37-59 in SI for detail.

8. Specify the number of test repetitions conducted for each vehicle. Additionally, indicate the sample sizes used for all calculations involving averages to enhance the transparency and reproducibility of the results.

Response:

Thanks for the comment. Table S1 in SI has been supplemented the number of test repetitions.

**"Table S1. Sets of test cycles.**

| No. | Vehicle ID | Ambient Temperature | Cold- or Hot-start Cycle | Repetitions |
|-----|-----------|---------------------|--------------------------|-------------|
| 1 | D1 | 23°C | Cold-start cycle | 2 |
| 2 | D1 | 23°C | Hot-start cycle | 2 |
| 3 | D2 | 23°C | Cold-start cycle | 2 |
| 4 | D2 | 23°C | Hot-start cycle | 2 |
| 5 | D2 | 0°C | Hot-start cycle | 2 |
| 6 | D3 | 23°C | Cold-start cycle | 3 |
| 7 | D3 | 23°C | Hot-start cycle | 3 |
| 8 | D4 | 23°C | Cold-start cycle | 3 |
| 9 | D4 | 23°C | Hot-start cycle | 3 |
| 10 | D4 | 0°C | Hot-start cycle | 2 |

"

Please refer to line 61 in SI for detail.

9. In Line 237-238: "The EF ratios across different volatility bins decreased with decreasing volatility, highlighting that the elevated I/SVOC EFs of HMVs were primarily due to a marked increase in organics within the volatility range of bins 2 to 6.". But according to the Fig. 3, the I/SVOCs EFs of HMVs and LMVs exhibited a rebound within the volatility range of bins 2 to 4. Is there any explanation for this phenomenon?

Response:

In fact, a similar volatility distribution phenomenon has also been found in previous studies on vehicle exhaust I/SVOC emissions. For example, both Zhao et al. (2015) and He et al. (2022) tested the exhaust from diesel vehicles and found an I/SVOC EF rebound around bin 2, as shown in the figure below, but they did not explain such a phenomenon. Liang et al. (2022) compared the I/SVOC volatility distribution in exhaust, diesel, and lubrication oil from an engine, using the Positive Matrix Factorization (PMF), and concluded that the EF rebound around bin 2 was attributed to the lubrication oil. However, we did not analyze the diesel and lubrication oil used in our test vehicles by TD-GC×GC-MS and without other evidence. We will improve in our future research.

[Figure]

Figure 4. I/SVOC volatility distribution in the study of Zhao et al. (2015) and He et al. (2022).

10. Around line 240, there's a comparison of the proportions of HMV and LMV organic
compounds; was there also a comparison of different component emission factors (EFs)
between HMV and LMV? Are all substances higher in HMV?

Response:

Thanks for the comment. The EFs of all organic compounds emitted by HMVs are higher than that
of LMVs, but the magnitude of the increase varies. The supplementary figure below has been added
to the SI and the main text has been modified as follows:

"

[Figure]

Figure S8. The average organic group distribution of HMVs and LMVs."

"…To further compare volatility and category distribution, the average EFs of HMVs and LMVs
are shown separately in Fig. 2. The EF ratios across different volatility bins exhibited a decreasing
trend with decreasing volatility, indicating that the elevated I/SVOC EFs of HMVs were primarily
due to a marked increase in organics within the volatility range of bins 2 to 6. Figure 2 further
depicts the relative proportion of distinct organic groups present in I/SVOC emissions and their
EFs are shown in Fig. S8. The EFs of all organic compounds emitted by HMVs were higher than
those of LMVs, but the magnitude of the increase varied. Except for phenol, alkene, and
cycloalkane, the organic group with the highest HMV-LMV ratio was carbonyls, up to 34, as shown in Fig. S8. The next highest is oxy-PAH & oxy-benzene, whose HMV-LMV ratio reached 11. The ratios of PAH_2rings, alcohol, and alkane were 7. Overall, the HMV-LMV ratios of O-I/SVOCs were relatively higher, which contributed 65% of the I/SVOCs emissions from HMVs, compared to 42% for LMVs. Since the SOA yields of O-I/SVOCs are lower than those of hydrocarbon-like I/SVOCs in the same bin (Chacon-Madrid and Donahue, 2011), variations in O-I/SVOC proportions directly impacted the SOAFP gap between HMVs and LMVs, which would be further discussed in Sect. 3.5. Alkane and oxy-PAH & oxy-benzene were the dominant contributors to I/SVOCs for both HMVs and LMVs. PAH_3rings contributed 8% of the I/SVOC emissions for HMVs, but 23% for LMVs. Interestingly, phenol, alkene, and cycloalkane were not detected in any of the LMV samples."

Please refer to line 238-242 in the main text and line 88 in SI for details.

11. The introduction and the section on SOA prediction would benefit from additional supporting references. Consider including following studies that explore the generation and sources of urban particulate matter to provide a more robust foundation for your discussion.

● Jacob M. Sommers, Craig A. Stroud, Max G. Adam, Jason O'Brien, Jeffrey R. Brook, Katherine Hayden, Alex K. Y. Lee, Kun Li, John Liggio, Cristian Mihele, Richard L. Mittermeier, Robin G. Stevens, Mengistu Wolde, Andreas Zuend, Patrick L. Hayes (2022) Evaluating SOA formation from different sources of semi- and intermediate-volatility organic compounds from the Athabasca oil sands. Environmental Science: Atmospheres. DOI: 10.1039/d1ea00053e.

● Qingsong Wang; Juntao Huo; Hui Chen*; Yusen Duan; Qingyan Fu; Yi Sun; Kun Zhang; Ling Huang; Yangjun Wang; Jiani Tan; Li Li*; Lina Wang; Dan Li; Christian George; Abdelwahid Mellouki, &Jianmin Chen (2023) Traffic, marine ships and nucleation as the main sources of ultrafine particles in suburban Shanghai, China. Environmental Science: Atmospheres. DOI: 10.1039/d3ea00096f.

● Ling Huang, Hanqing Liu, Greg Yarwood, Gary Wilson, Jun Tao, Zhiwei Han, Dongsheng Ji, Yangjun Wang, Li Li*. Modeling of secondary organic aerosols (SOA) based on two commonly used air quality models in China: Consistent S/IVOCs contribution but large differences in SOA aging. Science of the Total Environment 2023, 903, 166162. https://doi.org/10.1016/j.scitotenv.2023.166162.

● Yangjun Wang; Miao Ning; Qingfang Su; Lijuan Wang*; Sen Jiang; Yueyi Feng; Weiling Wu; Qian Tang; Shiyu Hou; Jinting Bian; Ling Huang; Guibin Lu; Kasemsan Manomaiphiboon; Burcak Kaynak; Kun Zhang; Hui Chen, &Li Li* (2024) Designing regional joint prevention and control schemes of PM2.5 based on source apportionment of chemical transport model: A case study of a heavy pollution episode. Journal of Cleaner Production. DOI: 10.1016/j.jclepro.2024.142313.

● Sahir Azmi, Mukesh Sharma (2023) Global PM$_{2.5}$ and secondary organic aerosols (SOA) levels with sectorial contribution to anthropogenic and biogenic SOA formation. Chemosphere. https://doi.org/10.1016/j.chemosphere.2023.139195.

Response:

Thanks for the suggestion. We have included these up-to-date relevant papers for reference and the main text has been revised as follows:

"As a major air pollutant, fine particulate matter (PM$_{2.5}$) leads to over three million premature deaths globally each year (Apte et al., 2018), mainly associated with lung cancer, ischemic heart disease, and stroke (Guan et al., 2018; Xue et al., 2021). Secondary organic aerosol (SOA) accounts for 12% to 77% of the total PM$_{2.5}$ mass based on global source apportionment results (Huang et al., 2014; Sun et al., 2020; Zhang et al., 2021). Observation studies have demonstrated that SOA

contributions increase with the severity of pollution during haze episodes in megacities in China (He et al., 2020; Ho, 2016; Li et al., 2015; Azmi et al., 2023; Wang et al., 2023; Wang et al., 2024). Among potential SOA precursors, intermediate-volatility and semi-volatile organic compounds (I/SVOCs), with effective saturation concentrations ($C^*$) between $10^3$ to $10^6$ and $10^0$ to $10^2$ $\mu g \cdot m^{-3}$, have been demonstrated to be more effective than volatile organic compounds (VOCs) (Daniel S. Tkacik et al., 2012; Jathar et al., 2013; Morino et al., 2022; Sommers et al., 2022; Huang et al., 2023). …"

"**Reference** …

Azmi, S. and Sharma, M.: Global PM2.5 and secondary organic aerosols (SOA) levels with sectorial contribution to anthropogenic and biogenic SOA formation, Chemosphere, 336, 139195, https://doi.org/10.1016/j.chemosphere.2023.139195, 2023.

…

Huang, L., Liu, H., Yarwood, G., Wilson, G., Tao, J., Han, Z., Ji, D., Wang, Y., and Li, L.: Modeling of secondary organic aerosols (SOA) based on two commonly used air quality models in China: Consistent S/IVOCs contribution but large differences in SOA aging, Sci Total Environ, 903, 166162, https://doi.org/10.1016/j.scitotenv.2023.166162, 2023.

…

Sommers, J. M., Stroud, C. A., Adam, M. G., O'Brien, J., Brook, J. R., Hayden, K., Lee, A. K. Y., Li, K., Liggio, J., Mihele, C., Mittermeier, R. L., Stevens, R. G., Wolde, M., Zuend, A., and Hayes, P. L.: Evaluating SOA formation from different sources of semi- and intermediate-volatility organic compounds from the Athabasca oil sands, Environ. Sci.: Atmos., 2, 469–490, https://doi.org/10.1039/D1EA00053E, 2022.

…

Wang, Q., Huo, J., Chen, H., Duan, Y., Fu, Q., Sun, Y., Zhang, K., Huang, L., Wang, Y., Tan, J., Li, L., Wang, L., Li, D., George, C., Mellouki, A., and Chen, J.: Traffic, marine ships and nucleation as the main sources of ultrafine particles in suburban Shanghai, China, Environ. Sci.: Atmos., 3, 1805–1819, https://doi.org/10.1039/D3EA00096F, 2023.

Wang, Y., Ning, M., Su, Q., Wang, L., Jiang, S., Feng, Y., Wu, W., Tang, Q., Hou, S., Bian, J., Huang, L., Lu, G., Manomaiphiboon, K., Kaynak, B., Zhang, K., Chen, H., and Li, L.: Designing regional joint prevention and control schemes of PM2.5 based on source apportionment of chemical transport model: A case study of a heavy pollution episode, Journal of Cleaner Production, 455, 142313, https://doi.org/10.1016/j.jclepro.2024.142313, 2024.

…"

Please refer to line 38-41, 370, 421, 460, and 474-480 in main text for details.

*Reference:*

*He, X., Zheng, X., You, Y., Zhang, S., Zhao, B., Wang, X., Huang, G., Chen, T., Cao, Y., He, L., Chang,*
*X., Wang, S., and Wu, Y.: Comprehensive chemical characterization of gaseous I/SVOC emissions from*
*heavy-duty diesel vehicles using two-dimensional gas chromatography time-of-flight mass spectrometry,*
*Environ. Pollut., 305, 119284, https://doi.org/10.1016/j.envpol.2022.119284, 2022b.*

*Liang, Z., Yu, Z., and Chen, L.: Quantifying the contributions of diesel fuel and lubricating oil to the*
*SVOC emissions from a diesel engine using GC × GC-ToFMS, Fuel, 310, 122409,*
*https://doi.org/10.1016/j.fuel.2021.122409, 2022.*

*Zhang, X., He, X., Cao, Y., Chen, T., Zheng, X., Zhang, S., and Wu, Y.: Comprehensive characterization*
*of speciated volatile organic compounds (VOCs), gas-phase and particle-phase intermediate- and semi-*
*volatile volatility organic compounds (I/S-VOCs) from Chinese diesel trucks, Sci. Total Environ., 912,*
*168950, https://doi.org/10.1016/j.scitotenv.2023.168950, 2024.*

*Zhao, Y., Nguyen, N. T., Presto, A. A., Hennigan, C. J., May, A. A., and Robinson, A. L.: Intermediate*
*volatility organic compound emissions from on-road diesel vehicles: chemical composition, emission*
*factors, and estimated secondary organic aerosol production, Environ. Sci. Technol., 49, 11516–11526,*
*https://doi.org/10.1021/acs.est.5b02841, 2015.*